# A novel approach to dry weight adjustments for dialysis patients using machine learning

Hae Ri Kim[1,2], Hong Jin Bae[3], Jae Wan Jeon[1], Young Rok Ham[2,4], Ki Ryang Na[2,4], Kang Wook Lee[2,4], Yun Kyong Hyon[5]‡*, Dae Eun Choi[2,4]‡*

**1** Division of Nephrology, Department of Internal Medicine, Chungnam National University Sejong Hospital, Sejong, South Korea, **2** Department of Nephrology, Medical School, Chungnam National University, Daejeon, South Korea, **3** Division of Nephrology, Department of Internal Medicine, Cheongju St. Mary's Hospital, Cheongju, South Korea, **4** Department of Medical Science, Medical School, Chungnam National University, Daejeon, South Korea, **5** Medical Mathematics Division, National Institute for Mathematical Sciences, Daejeon, South Korea

☯ These authors contributed equally to this work.
‡ These authors also contributed equally to this work.
* hyon@nims.re.kr (YKH); daenii@cnu.ac.kr (DEC)

**Data Availability Statement:** Data cannot be shared publicly because data contains potentially identifying or sensitive patient information. Data are available from the Institutional Data Access / Ethics Committee of Chungnam National University

## Abstract

### Background and aims

Knowledge of the proper dry weight plays a critical role in the efficiency of dialysis and the survival of hemodialysis patients. Recently, bioimpedance spectroscopy(BIS) has been widely used for set dry weight in hemodialysis patients. However, BIS is often misrepresented in clinical healthy weight. In this study, we tried to predict the clinically proper dry weight ($DW_{CP}$) using machine learning for patient's clinical information including BIS. We then analyze the factors that influence the prediction of the clinical dry weight.

### Methods

As a retrospective, single center study, data of 1672 hemodialysis patients were reviewed. $DW_{CP}$ data were collected when the dry weight was measured using the BIS ($DW_{BIS}$). The gap between the two ($Gap_{DW}$) was calculated and then grouped and analyzed based on gaps of 1 kg and 2 kg.

### Results

Based on the gap between $DW_{BIS}$ and $DW_{CP}$, 972, 303, and 384 patients were placed in groups with gaps of <1 kg, ≧1kg and <2 kg, and ≧2 kg, respectively. For less than 1 kg and 2 kg of GapDW, It can be seen that the average accuracies for the two groups are 83% and 72%, respectively, in usign XGBoost machine learning. As $Gap_{DW}$ increases, it is more difficult to predict the target property. As $Gap_{DW}$ increase, the mean values of hemoglobin, total protein, serum albumin, creatinine, phosphorus, potassium, and the fat tissue index tended to decrease. However, the height, total body water, extracellular water (ECW), and ECW to intracellular water ratio tended to increase.

Hospital (contact via irb@cnuh.co.kr) for researchers who meet the criteria for access to confidential data.

**Funding:** This research was supported by National Institute for Mathematical Sciences (NIMS) grant funded by the Korea government, 2021 (No. NIMS-B21910000) and Basic Science Research Program through the National Research Foundation of Korea (NRF) funded by the Ministry of Education (NRF-2017R1D1AB03035061). However, the funders had no role in study design, data collection and analysis, decision to publish, or preparation of the manuscript.

**Competing interests:** The authors have declared that no competing interests exist.

## Conclusions

Machine learning made it slightly easier to predict $DW_{CP}$ based on $DW_{BIS}$ under limited conditions and gave better insights into predicting $DW_{CP}$. Malnutrition-related factors and ECW were important in reflecting the differences between $DW_{BIS}$ and $DW_{CP}$.

## Introduction

Accurately establishing dry weight is important in patients with end stage renal disease (ESRD) on hemodialysis. It is well known that identifying the proper dry weight is associated with the adequate efficiency of dialysis, minimizes the burden on the cardiovascular system, and improves the survival rate of dialysis patients [1–3].

In general, dry weight is(DW) defined as the lowest tolerated postdialysis weight achieved via gradual change in postdialysis weight at which there are minimal signs or symptoms of hypovolemia or hypervolemia [4].

It has been reported that DW overestimation is a major risk factor in the development of hypertension, left ventricular hypertrophy, and cardiovascular disease, therefore affecting the mortality risk of patients on hemodialysis [5–8]. Underestimation of DW can result in hypovolemia and can induce hypotension, cramps, and dizziness [9,10]. Hypovolemia can also result in a reduced blood supply to vital organs, causing ischemia and consequently contributing to the loss of residual renal function [8]. As a consequence, it is crucial to implement an efficient and precise method to determine DW in patients with ESRD on hemodialysis.

DW assessment via clinical signs, such as measurements of blood pressure, jugular venous pressure, and the presence or absence of edema, is the most commonly used method. However, this method does not reflect the patient's underlying illnesses and decreases in muscle mass [11].

Over the years, paraclinical tools have been developed to help nephrologists assess DW. Measurements of the cardiothoracic ratio on chest X-rays has long been used but is imprecise due to its dependency on underlying cardiovascular conditions, such as cardiac hypertrophy [12]. Ultrasonic measurements of the inferior vena cava diameter and lung can also be useful tools for measuring DW but are limited because they only provide information concerning overhydration [13,14].

Recently, a method to determine the amount of body water using the bioimpedance spectroscopy (BIS) method has been developed [15]. Even though DW measurements using BIS are useful and reliable when applied to actual clinical patients, there have been reports that the measured DW was not optimal for some patients depending on the prescription of blood pressure medications and the presence of cardiovascular disease [7,16].

Recently, machine-learning-based artificial intelligence has been successfully applied in the medical field and has shown promising results in predicting complications [17–19]. Machine-learning approaches can be classified into two main types: unsupervised learning for unlabeled data and supervised learning for labeled data. The latter is more appropriate for predicting complications; decision tree-based machine-learning algorithms are especially appropriate for table-type data with labels, i.e., target factors [20–22]. However, there are few studies that use machine-learning approaches to predict DW in hemodialysis patients.

In this study, machine learning was applied to accurately determine DW. Using this approach, various clinical factors influencing patients with a large gap between the DW measured using BIS ($DW_{BIS}$) and the clinically stable dry weight were identified and analyzed.

Additionally, we evaluated whether a more accurate clinical dry weight could be presented using machine learning based on the various clinical factors and BIS data of the patient.

## Materials and methods

### Study design

The present study was approved by the institutional review board of Chungnam National University Hospital (IRB number: CNUH 2015-12-025-003). This study was entirely retrospective (using existing data and medical records available as of the date of submission of the IRB), and was permitted a consent waiver through the IRB. The study was conducted as a single center and retrospective study on dialysis patients who were treated at the Chungnam National University Hospital between January 2011 and September 2015. All data were collected from the medical records of the Chungnam National University Hospital. Information on mortality was obtained from the database of the National Health Insurance Service.

### Patient selection

All patients were diagnosed with ESRD and started maintenance hemodialysis between June 2011 and December 2015. Maintenance hemodialysis is defined here as hemodialysis performed three times a week.

Dialysis was maintained for at least 3 months after initiation. Patients who received temporarily hemodialysis due to acute renal injury were excluded. In addition, we excluded patients with unstable clinical conditions (i.e., acute infections or intensive care unit admission). Patients for whom we did not have information concerning their hydration status or who's DW could not be confirmed using the BIS method were excluded. Patients whose antihypertensive drug prescriptions changed between the 2 weeks before and after the time their body compositions were measured were also excluded.

### Body composition measurements

Body composition, including hydration status (overhydration; OH), was assessed using a portable BIS device (BCM; Fresenius Medical Care, Bad Homburg, Germany). Patient measurements were obtained on the day of dialysis prior to dialysis. All data measured using the BIS method, including extracellular water (ECW), intracellular water (ICW), total body water (TBW), lean tissue index (LTI), fat tissue index (FTI), ECW/ICW (E/I) ratio, body mass index (BMI), lean tissue mass (LTM), fat mass (FAT), adipose tissue mass (ATM), and body cell mass (BCM), were included. These fluid volumes were then used to determine the fluid overload, expressed as the OH value. OH was measured as a negative or positive value. ECW, ICW, TBW, and OH are all expressed in units of liters.

### Definitions

In general, DW is defined as the lowest tolerated postdialysis weight achieved via gradual change in postdialysis weight at which there are minimal signs or symptoms of hypovolemia or hypervolemia [4]. In this study, the concept of clinically proper DW (DW$_{CP}$) was used to distinguish it from DW$_{BIS}$. DW$_{CP}$ was defined as the lowest post-dialysis weight a patient could tolerate without signs or symptoms of overhydration or dehdyration during or after dialysis. DW was defined by combining judgments on the clinical situation with reference to DW predicted by the BIS method. The clinical judgment of the appropriate dry weight used the patient's indications and imaging studies. At the center where the study was conducted, the medical staff checks and records events from the previous dialysis to the day on the dialysis

day. For example, when determining overhydration, it was included based on cases where peripheral edema, generalized edema or chest discomfort, or pleural effusion or pulmonary edema was not observed in chest PA. We checked whether musle spasm or dizziness occurred through medical records. The OH value measured using BIS is denoted $OH_{BIS}$, and the DW assessed using BIS ($DW_{BIS}$) was calculated from the body weight (Weight just before dialysis) minus the $OH_{BIS}$ value (kg). The gap between $DW_{CP}$ and the body weight (Weight just before dialysis) is denoted $OH_{CP}$. The measurement gap between $DW_{BIS}$ and $DW_{CP}$ is the DW gap ($Gap_{DW}$), i.e., it is defined as the absolute value of the body weight minus $DW_{BIS}$ from $DW_{CP}$ in units of kg.

## Measurements of clinical parameters

The baseline patient characteristics, its mean and standard deviation (SD) analyzed included the age on the start day of dialysis (days), sex, height (cm), initial body weight (kg), predialysis systolic/diastolic blood pressure (mmHg), and comorbidities (presence of diabetes or hypertension). Factors predicted to affect the patient's DW were collected from the blood test results. Laboratory tests were performed within three days before and after the $DW_{BIS}$ measurement, and hemoglobin, blood urea nitrogen, creatinine (Cr), albumin, total calcium (TCa), phosphorus (P), sodium (Na), potassium (K), and chloride (Cl) measurements were included.

## Machine-learning adjustment

The machine-learning methodology, which is currently attracting attention, was applied to predict OH (proper). Since many external sources on machine learning exist, detailed information on the introduction and methodology of machine learning is not included. Briefly, machine learning is a very effective method of learning a given data to predict a specific aim attribute (OHcp in this paper), and using it to predict the corresponding aim attribute when new patients data are inputted. A lot of research is being conducted, and it is widely used in artificial intelligence technology. In this study, we used a decision tree-based methodology suitable for application to table-type medical data, the light gradient boosting method (LightGBM). the extreme gradient boosting tree (XGBoost), and random forest methods [20–22]. These tree-based methods are more efficient for table type data structure and relatively small data set, especially, RF is better than others for given small data set. Even though deep learning is more famous in machine learning applications, it requires bigger data set for learning model and getting certain prediction accuracy. Before learning the prediction model from the collected data, outliers were first removed by pre-processing the data. To verify the prediction result, 20 random samplings were performed. In each random sample, 80% of the collected data were sampled and used for training and the remaining 20% were used for testing. The suggested values obtained from the predicted results are presented after taking the average of the 20 samplings.

## Results

### Baseline characteristics

The average age of the patients was 65.1 years. There were 672 (40%) men, and diabetes and hypertension were found in 1,002 (50%) and 1,328 (79%) of the patients, respectively. The average duration of dialysis was approximately 806 days.

To confirm the predicted level of machine learning according to the gap between $DW_{BIS}$ and $DW_{CP}$, the analysis was divided into groups with differences of less than 1 kg and of more than 2 kg. In patients with a $Gap_{DW}$ value of more than 2 kg, the proportion of women was high, the age was high, and there were many patients with diabetes (Table 1).

## Prediction of clinical DW using machine learning

The accuracy of the results when applying machine learning to all the collected data is shown in Table 2. As can be seen from the results, the prediction accuracy of machine learning for $OH_{CP}$ was very low, less than 40%, and the maximum error for $OH_{CP}$ was very high at 1.5 kg. It was difficult to obtain a valid machine-learning-based predictive model, which is required for $OH_{CP}$ prediction.

We classified and extracted two groups based on $Gap_{DW}$ ($|DWCP-DW_{BIS}|$) from the collected data. One group contained data where $Gap_{DW}$ is less than 1 kg, and the other contained

**Table 1. Baseline characteristics.**

| Group | Total N = 1,672 | | GAP (<1 kg) N = 972 | | GAP (≧1kg, <2 kg) N = 303 | | GAP (>2 kg) N = 384 | |
|---|---|---|---|---|---|---|---|---|
| Feature | Mean | SD | Mean | SD | Mean | SD | Mean | SD |
| Age | 65.01 | 12.23 | 64.91 | 12.44 | 65.56 | 11.58 | 64.84 | 12.30 |
| Gender | 672 (40%) | 0.49 | 408 (42%) | 0.49 | 119 (39%) | 0.49 | 142 (37%) | 0.48 |
| Height (cm) | 161.46 | 8.51 | 160.98 | 8.25 | 161.14 | 8.62 | 162.69 | 8.92 |
| Weight (kg) | 60.50 | 11.30 | 60.24 | 11.24 | 60.34 | 10.39 | 61.15 | 12.07 |
| Vintage of HD (day) | 806.79 | 826.20 | 786.98 | 809.59 | 837.01 | 808.27 | 827.89 | 873.59 |
| DM | 1,002 (50%) | 0.49 | 561 (58%) | 0.49 | 185 (61%) | 0.49 | 247 (64%) | 0.48 |
| HTN | 1,328 (79%) | 0.40 | 797 (82%) | 0.38 | 240 (79%) | 0.41 | 279 (73%) | 0.45 |
| Hb (g/dL) | 10.13 | 1.55 | 10.25 | 1.50 | 10.15 | 1.59 | 9.83 | 1.61 |
| Total protein (g/dL) | 6.34 | 0.70 | 6.37 | 0.66 | 6.43 | 0.64 | 6.19 | 0.81 |
| Albumin (g/dL) | 3.38 | 0.57 | 3.46 | 0.55 | 3.39 | 0.49 | 3.17 | 0.65 |
| BUN (mg/dL) | 56.05 | 23.65 | 57.72 | 23.21 | 53.94 | 23.29 | 52.95 | 24.07 |
| Cr (mg/dL) | 7.68 | 3.37 | 7.98 | 3.26 | 7.73 | 3.35 | 6.89 | 3.55 |
| Tca (mg/dL) | 8.36 | 0.79 | 8.36 | 0.77 | 8.33 | 0.73 | 8.37 | 0.88 |
| P (mg/dL) | 4.35 | 1.57 | 4.43 | 1.55 | 4.41 | 1.66 | 4.10 | 1.48 |
| Na (mEq/L) | 137.06 | 3.97 | 137.18 | 3.95 | 136.70 | 3.71 | 137.03 | 4.15 |
| K (mEq/L) | 4.74 | 0.92 | 4.77 | 0.90 | 4.81 | 0.96 | 4.60 | 0.96 |
| Cl (mEq/L) | 101.34 | 5.16 | 101.59 | 5.16 | 100.69 | 5.06 | 101.21 | 5.13 |
| TBW (L) | 32.00 | 7.11 | 31.62 | 6.90 | 32.08 | 6.81 | 32.85 | 7.79 |
| ECW (L) | 15.54 | 3.51 | 15.19 | 3.32 | 15.59 | 3.23 | 16.37 | 3.99 |
| ICW (L) | 16.45 | 4.20 | 16.41 | 4.03 | 16.48 | 4.08 | 16.51 | 4.71 |
| E/I | 0.97 | 0.19 | 0.94 | 0.16 | 0.97 | 0.18 | 1.03 | 0.24 |
| BMI (kg/m$^2$) | 23.16 | 3.80 | 23.18 | 3.78 | 23.20 | 3.52 | 23.09 | 4.06 |
| LTI (kg/m$^2$) | 13.05 | 3.42 | 13.07 | 3.24 | 13.18 | 3.32 | 12.92 | 3.92 |
| FTI (kg/m$^2$) | 9.11 | 4.61 | 9.23 | 4.56 | 8.96 | 4.52 | 8.88 | 4.83 |
| LTM (kg/m$^2$) | 34.39 | 10.74 | 34.31 | 10.34 | 34.59 | 10.60 | 34.41 | 11.85 |
| FAT (kg) | 18.70 | 9.67 | 18.74 | 9.48 | 18.29 | 9.04 | 18.71 | 10.52 |
| ATM (kg) | 21.85 | 10.79 | 22.08 | 10.56 | 21.34 | 10.50 | 21.57 | 11.64 |
| BCM (kg) | 18.93 | 7.28 | 18.92 | 6.99 | 19.13 | 7.18 | 18.82 | 8.08 |
| $DW_{BIS}$ (kg) | 58.41 | 11.19 | 58.44 | 10.88 | 58.12 | 10.32 | 58.35 | 12.54 |
| $OH_{BIS}$ (kg) | 2.11 | 2.40 | 1.77 | 1.91 | 2.17 | 2.07 | 2.94 | 3.32 |
| $DW_{CP}$ (kg) | 58.63 | 10.94 | 58.44 | 10.86 | 58.44 | 10.16 | 59.29 | 11.50 |
| $Gap_{DW}$ (kg) | 0.22 | 3.07 | 0.00 | 0.48 | 0.32 | 1.50 | 0.94 | 4.79 |
| $OH_{CP}$ (kg) | 1.87 | 3.17 | 1.80 | 2.15 | 1.91 | 2.07 | 1.87 | 4.00 |

Abbreviations: SD, standard deviation; HD, Hemodialysis; DM, diabetes mellitus; HTN, hypertension; Hb, hemoglobin; BUN, blood urea nitrogen; Cr, serum creatinine; TCa, serum total calcium; P, serum phosphorus; Na, sodium; K, serum potassium; Cl, serum chloride; TBW, total body water; ECW, extracellular water; ICW, intracellular water; E/I, extracellular water to intracellular water ratio; BMI, body mass index; LTI, lean tissue index; FTI, fat tissue index; lean tissue mass, LTM; FAT, fat mass; ATM, adipose tissue mass; BCM, body cell mass; DW, dry weight; OH, overhydration.

**Table 2. Prediction using machine learning.**

| | | LightGBM Test Accuracy (%) (OH-CP (MAE)) | XGBoost Test Accuracy (%) (OH-CP (MAE)) | Random Forest Test Accuracy (%) (OH-CP (MAE)) |
|---|---|---|---|---|
| $Gap_{DW} < 1$ kg | Mean | 81.25% (547.0 g) | 82.89% (515.5 g) | 79.52% (570.3 g) |
| | Max | 84.57% (499.1 g) | 86.33% (459.9 g) | 83.11% (530.1 g) |
| | Min | 73.85% (622.1 g) | 75.21% (596.1 g) | 70.44% (656.2 g) |
| $Gap_{DW} < 2$ kg | Mean | 69.71% (770.6 g) | 72.02% (734.6 g) | 71.00% (747.3 g) |
| | Max | 78.33% (693.9 g) | 80.06% (674.3 g) | 79.03% (682.2 g) |
| | Min | 53.76% (876.9 g) | 55.84% (831.0 g) | 54.18% (857.0 g) |
| Total | Mean | 23.58% (1,351.9 g) | 28.54% (1,287.5 g) | 30.82% (1,250.5 g) |
| | Max | 35.85% (1,224.6 g) | 38.83% (1,178.8 g) | 39.36% (1,139.4 g) |
| | Min | 11.19% (1,448.7 g) | 21.01% (1,404.2 g) | 21.43% (1,377.9 g) |

All collected data (1,672 patients) were included. CP(MAE) indicates the clinically proper mean absolute error.

data where $Gap_{DW}$ is less than 2 kg. The results of applying machine learning to these two groups are presented in Table 2. It can be seen that the average accuracies for the two groups are 83% and 72%, respectively, in the case of XGBoost; in particular, the accuracy for the $Gap_{DW} < 1$ kg patients was dramatically improved compared to the prediction result for the entire dataset. It is obvious from this result that, when data with large $Gap_{DW}$ are included and as $Gap_{DW}$ increases, it is increasingly difficult to predict the target property ($OH_{CP}$) as a BIS measurement property.

In terms of machine learning, the difference between the two groups can also be confirmed via the feature importance of the machine-learning methodology. Comparing the feature importance of the two groups (Group 1: $Gap_{DW} < 1$ kg and Group 2: $Gap_{DW} < 2$ kg) reveals very different patterns, as shown in Fig 1. In the feature importance of Group 1, E/I and ECW (L) play important roles in learning the machine-learning predictive model and predicting the target feature; the other features, especially TBW (L), appear to play a secondary role. Conversely, Group 2, which includes patients with $Gap_{DW}$ between 1 kg and 2 kg, shows a further enhancement in the importance of E/I and ECW (L). In addition, the roles of features of low importance in Group 1 are slightly increased in Group 2 and distributed downward. Therefore, the data (1 kg $\leqq Gap_{DW} < 2$ kg) newly included in Group 2 changes the properties of the dataset and appears to play a role in lowering the accuracy of the prediction.

## The factors affecting $Gap_{DW}$

To compare the clinical differences from the point of view of the data distribution, box plots were used to compare the clinical factors in groups with $Gap_{DW}$ less than 1 kg, between 1 kg and 2 kg, and greater than 2 kg (Fig 2). In box plot, the mean values of hemoglobin, total protein, serum albumin, serum creatinine, phosphorus, serum sodium, serum potassium, and FTI tended to decrease, as the $Gap_{DW}$ increased. The height, TBW, ECW, and E/I ratio tended to increase as the $Gap_{DW}$ increased. LTI and BMI showed no change in their trends for the three groups.

## Discussion

In this study, we derived a more clinically accurate DW using machine learning based on volume status information measured using BIS as well as clinical information concerning the patients.

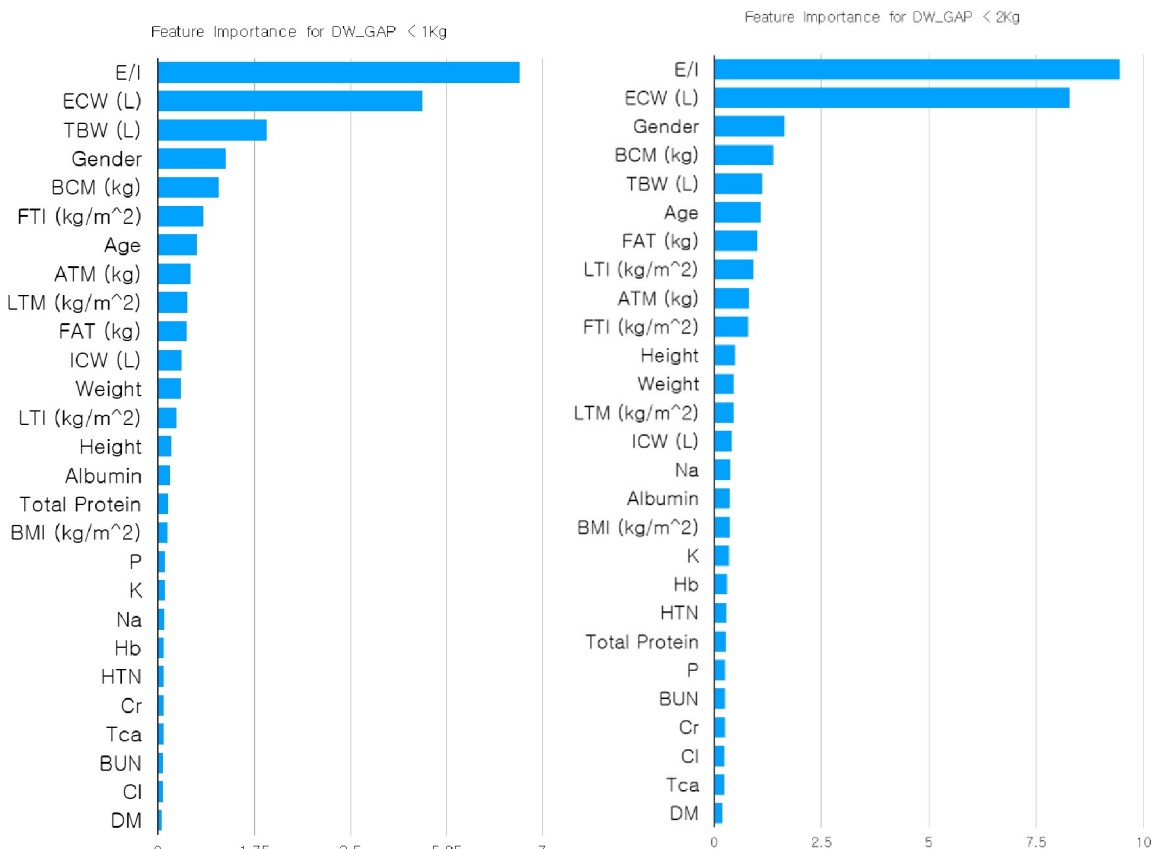

**Fig 1. Feature importance for the two groups (Group 1: Gap$_{DW}$ < 1 kg and Group 2: Gap$_{DW}$ < 2 kg).**

In addition, it was confirmed that patient's blood chemistry, including hemoglobin, total protein, albumin, creatinine, phosphorus, and potassium, was a factor that caused a gap between DW$_{BIS}$ and DW$_{CP}$.

It is important to maintain adequate hydration during the treatment of patients with ESRD on hemodialysis. Determining the amount and rate of water removal by targeting the appropriate patient DW during dialysis treatment is a major part of the dialysis regimen. In general, the DW evaluation is determined by a clinical decision. DW is defined as the lowest weight that is tolerable to a patient without symptoms or the occurrence of hypotension [23].

Several methods have been proposed to measure the volume status, and blood tests including atrial or brain natriuretic peptide levels, inferior vena cava diameter measurements using ultrasound, and blood pressure measurements have been used. However, DW results using these tools are not accurate [24,25]. BIS or multifrequency bioimpedance analysis was introduced to establish DW in dialysis patients [15]. Using a method to distinguish between ECW and ICW over multiple frequencies, it is possible to more accurately predict DW by measuring the volume status of a dialysis patient [26]. However, sometimes, dialysis targeting DW using the BIS method can result in a hypotensive status or peripheral and pulmonary edema in dialysis patients [16].

In this study, we tried to correct the gap between DW$_{BIS}$ and DW$_{CP}$ using various clinical data gathered from patients. The application of machine learning to the BIS-based data and clinical parameters resulted in accuracy compatible with that of predicting OH$_{CP}$ using DW$_{BIS}$ alone; however, the mean absolute error (MAE) for OH$_{CP}$ was improved.

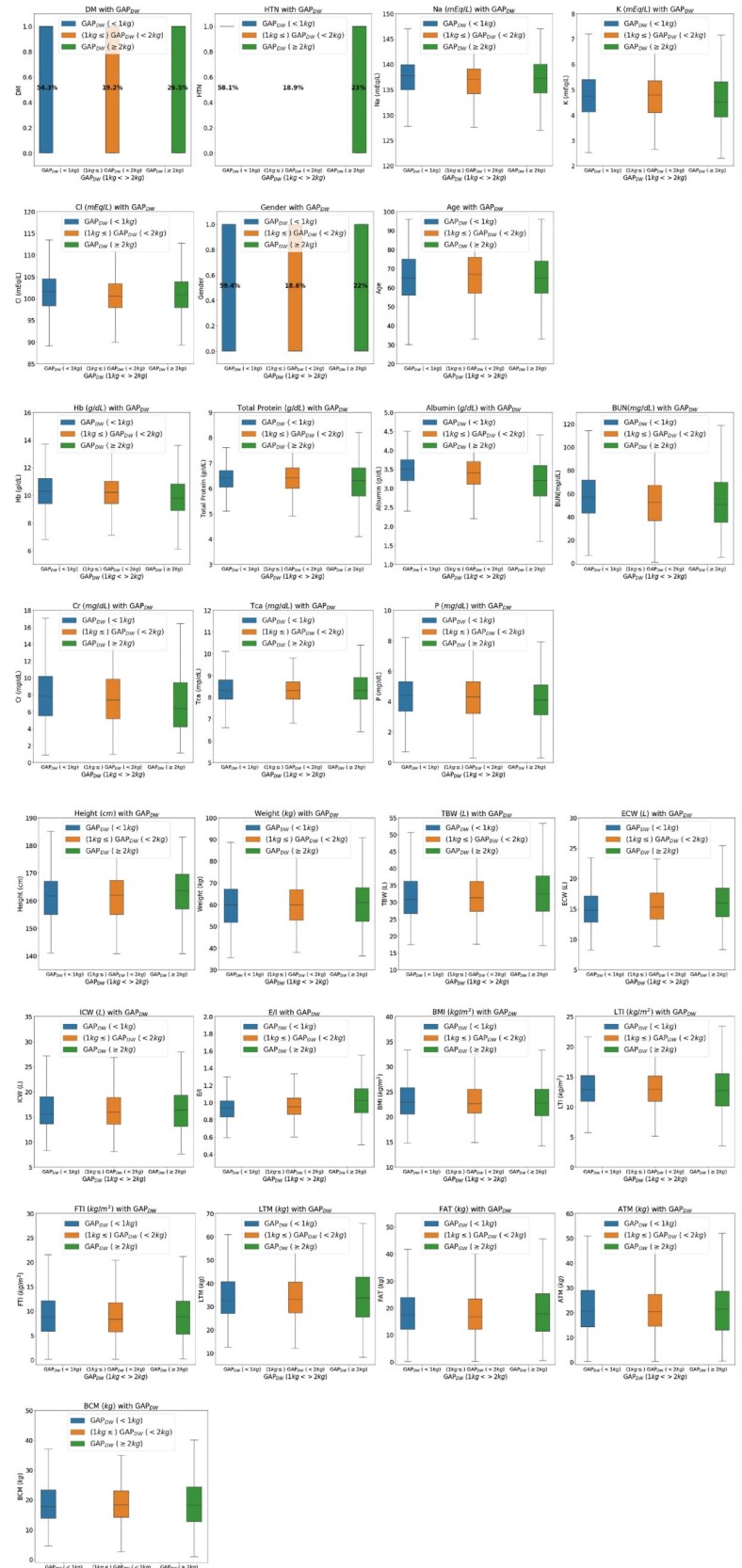

**Fig 2.** Distribution of the data according to GapDW: (A) clinical data; (B) blood chemistry; and (C) BIS parameters.

In particular, when only data with a $Gap_{DW}$ value of less than 1 kg were analyzed as a data-set using machine learning, a mean accuracy of 83.26% and a maximum accuracy of 86.65% were found using the XGBoost method and the predicted difference from $OH_{CP}$ was up to 458 ml. For $OH_{CP}$, the average measurement range was 731 ml and it was more predictable for $DW_{CP}$ than for BIS-based data alone. When using all of the data, the method was very inaccurate when predicting $OH_{CP}$; however, because BIS-based data play an important role in the prediction, when there was too much of a difference with respect to the clinical DW, or when data that accurately reflect the volume status were used, the machine-learning prediction was thought to be inaccurate overall.

In view of the $Gap_{DW}$ groups, the box plots indicated that the $Gap_{DW}$ difference and older ages did not show a tendency but that the importance of age increased in the group with a $Gap_{DW}$ difference of more than 2 kg. In a previous study, when comparing the hydration status of elderly dialysis patients and young dialysis patients, it was reported that ICW is low, ECW/TBW tends to be low, and a larger hydration status is shown. In the distribution of the actual data, an age trend was not observed but it was confirmed that the importance of the features analyzed by machine learning is useful for applications to actual clinical data.

Box plots were used to reveal the distributions of the BIS-based data and the clinical data according to the gap differences based on $Gap_{DW}$. In the box plots, the mean value of ECW showed a tendency to increase as $Gap_{DW}$ increased. Conversely, the overall distribution of hemoglobin, serum total protein, serum albumin, serum creatinine, phosphorus, and serum potassium had lower values as $Gap_{DW}$ increased.

The ECW value reflects overhydration. As $Gap_{DW}$ increases, ECW tends to increase. As the degree of overhydration increases, the prediction accuracy of $DW_{BIS}$ is expected to decrease.

Previous studies have shown that hypoalbuminemia (low serum albumin concentration) is associated with overhydration in patients undergoing maintenance dialysis. ECW increased in patients with low serum albumin concentrations, and similar results were shown regardless of the dialysis method. In addition, when hypoalbuminemia is present, overall nutritional weight, BMI, BUN, serum creatinine and potassium levels, and dietary protein intake are normal [27–29]. Hypophosphatemia is significantly associated with overhydration. In addition, it has been reported to be related to higher age and lower serum albumin, creatinine, hemoglobin, and serum calcium levels [30].

In previous studies, a relationship between overhydration and low hemoglobin concentration was reported and anemia tended to worsen when overhydration was severe. In addition to chronic kidney disease, more than half of patients with advanced heart failure who could show volume overload showed anemia and it was reported that this anemia was caused by a dilution effect due to fluid overload [31,32]. There are limitations to directly using anemia as a measure of overhydration; however, it can be considered as a factor affecting the increase in $Gap_{DW}$.

In several previous studies, serum creatinine was used as a biomarker for muscle metabolism when assessing muscle mass in patients with ESRD on dialysis. Early studies estimating muscle mass based on creatinine kinetics demonstrated good correlation with other estimates of muscle mass [33,34]. In addition, serum creatinine has been significantly correlated with LTM [35]. Based on these findings, lower predialysis serum creatinine levels are predicted to affect the $Gap_{DW}$ increase.

Hypokalemia is used as an indicator of malnutrition, and its association with malnutrition via albumin and total protein can be estimated [36]. Hypokalemia showed a tendency to drop significantly in the group with $Gap_{DW}$ of more than 2 kg. Based on these results, caution is needed when interpreting and applying $DW_{BIS}$ to patients with hypokalemia and/or malnutrition.

These blood chemistry findings may not be related to malnutrition. Even though there was no tendency in LTI and BMI according to the $Gap_{DW}$ increase in the box plots, these blood chemistry findings are associated with a decrease in the effective circulating volume and an increase in interstitial edema, consequently making the prediction of $DW_{CP}$ difficult.

There are some limitations to this study. First, the DW measurements using the BIS device were performed by a single person; therefore, there is a possibility that an error in the measurement result may be included. Second, because the amount of data for the group with $Gap_{DW}$ of more than 2 kg is small, it is not suitable for training a machine-learning predictive model for $OH_{CP}$. However, it is important to obtain a predictive model and proper factors other than the BIS factors to improve the treatment of patients in this group. Third, in this study, an analysis by gender and age was not performed. Such an analysis in the future will enable a more detailed approach for these patient groups. Fourth, it was not possible to evaluate whether the external cohort population showed the same pattern as a single-center study. Fifth, the patient's mortality and morbidity were not analyzed in relation to water control through more accurate measurement of dry weight. In future studies, prospective studies and multicenter studies will be conducted, and the long-term prognosis of patients will need to be analyzed.

In conclusion, we found the feature importance of groups according to differences in $Gap_{DW}$. It was confirmed that ECW and malnutrition-related blood chemistry findings, including hemoglobin, serum total protein, serum albumin, serum creatinine, serum phosphorus, and serum potassium, were important features. Machine learning made better predictions of $DW_{CP}$ based on $DW_{BIS}$ in patients with $Gap_{DW}$ of less than 2 kg. It is difficult to predict $DW_{CP}$ using machine learning for patients with $Gap_{DW}$ of more than 2 kg; therefore, it is necessary to analyze additional appropriate features for future extreme cases.

In the future, if more patients are included to increase the prediction accuracy using machine learning, this technique will be helpful in establishing the appropriate DW for patients. Machine-learning predictive models can be helpful to establish $aOH_{CP}$.

## Author Contributions

**Conceptualization:** Hae Ri Kim, Hong Jin Bae, Yun Kyong Hyon, Dae Eun Choi.

**Data curation:** Hae Ri Kim, Hong Jin Bae.

**Funding acquisition:** Yun Kyong Hyon, Dae Eun Choi.

**Methodology:** Hae Ri Kim, Hong Jin Bae, Jae Wan Jeon.

**Project administration:** Yun Kyong Hyon.

**Resources:** Hong Jin Bae, Jae Wan Jeon, Young Rok Ham, Dae Eun Choi.

**Software:** Young Rok Ham.

**Supervision:** Ki Ryang Na, Kang Wook Lee, Yun Kyong Hyon, Dae Eun Choi.

**Validation:** Jae Wan Jeon, Ki Ryang Na, Kang Wook Lee, Yun Kyong Hyon, Dae Eun Choi.

**Writing – original draft:** Hae Ri Kim.

**Writing – review & editing:** Yun Kyong Hyon, Dae Eun Choi.

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
