## [Decision Letter · Decision Letter 0]

10 Feb 2021

PONE-D-20-40081

A novel approach for adjustment of to dry weight in adjustments for dialysis patients using machine learning.

PLOS ONE

Dear Dr. Choi,

Thank you for submitting your manuscript to PLOS ONE. After careful consideration, we feel that it has merit but does not fully meet PLOS ONE’s publication criteria as it currently stands. Therefore, we invite you to submit a revised version of the manuscript that addresses the points raised during the review process.This is an interesting study as your team has tried a newer approach of dry weight in adjustments for dialysis patients using machine learning .After reviewing this paper and also the reviewer comments several issues needs to be addressed before this study can be considered for publication.Please address the comments  and concerns by the  reviewers  and after that your paper will be reviewed again before being considered for publication.

We look forward to receiving your revised manuscript.

Kind regards,

Bhagwan Dass, MD

Academic Editor

PLOS ONE

Journal Requirements:

2. To comply with PLOS ONE submission guidelines, in your Methods section, please provide additional information regarding the methodological and statistical analyses performed for the computational tests. For more information on PLOS ONE's expectations for statistical reporting, please see https://journals.plos.org/plosone/s/submission-guidelines.#loc-statistical-reporting.

5.Thank you for stating the following in the Acknowledgments Section of your manuscript:

"This research was also supported by National Institute for Mathematical Sciences (NIMS) grant funded

by the Korea government, 2020 (No. NIMS-2020B900000). This research was supported by Basic

Science Research Program through the National Research Foundation of Korea (NRF) funded by the

Ministry of Education (NRF-2017R1D1AB03035061)."

 "he funders had no role in study design, data collection and analysis, decision to publish, or preparation of the manuscript."

Reviewers' comments:

Reviewer's Responses to Questions

**Comments to the Author**

1. Is the manuscript technically sound, and do the data support the conclusions?

Reviewer #1: Yes

Reviewer #2: Partly

Reviewer #3: No

2. Has the statistical analysis been performed appropriately and rigorously? 

Reviewer #1: Yes

Reviewer #2: No

Reviewer #3: No

3. Have the authors made all data underlying the findings in their manuscript fully available?

Reviewer #1: Yes

Reviewer #2: No

Reviewer #3: No

4. Is the manuscript presented in an intelligible fashion and written in standard English?

Reviewer #1: Yes

Reviewer #2: No

Reviewer #3: No

5. Review Comments to the Author

Reviewer #1: Choi and coworkers are submitting a novel approach for adjusting dry weight in hemodialysis patients using machine learning support. For this purpose, they developed a retrospective study using a data set of 1672 hemodialysis (HD) patients in which they compared dry weight defined clinically (clinical proper) to dry weight predicted or estimated from bioimpedance spectroscopy device (BIS). They estimated the gap between these two methods and they clustered such difference in three categories (<1; 1-2;>2 liters). With machine learning support, they identified and weighted factors (clinical data, lab test, body composition from BIS) being associated with each of these groups. They concluded that machine learning support, may help to improve dry weight clinical estimate using BIS and additional factors (clinical data, lab values, body composition) particularly in the lowest fluid overload patient categories. In addition, malnutrition related factors may explain most of gap discrepancy in dry weight estimate.

This novel approach relying on machine learning and artificial intelligence is interesting, and seems appealing as support tool for guiding physicians in managing more precisely dialysis patients. This approach may be used as an example of featuring future of medicine.

Now the study raises several concerns that should be addressed:

1. Use of such machine learning and intelligence artificial tool to identify factors that may influence bad estimate of dry weight such malnutrition is not sufficient to validate this approach.

2. It is not clear from reading, how the authors defined the ideal dry weight of patients? Was it the fluid overload estimated from the BIS device that was chosen as target to set suitable or ideal dry weight or the clinical judgement or a combination? That should be defined clearly since it is confusing along the manuscript.

3. If the ideal or suitable dry weight was established on the clinical and/or biologic assessment, therefore the authors should define which criteria and threshold values were used to define the three categories of gap discrepancies.

4. If the target dry weight was established from BIS measurement then it is easier to understand but still clinical criteria used to define fluid status needs to be listed with their threshold values.

5. From a methodological point of view, it would be also interesting to validate the algorithm that was developed in the assessment of fluid overload within an external cohort of patients.

6. Clinical outcomes of these three categories of patients would require further analyses. This is important to value the support of machine learning in reducing intradialytic morbidity (ie, incidence of intradialytic hypotension) and/or improving medium- or long-term morbidity (ie, hospitalization for pulmonary edema) and mortality (all cause or cardiovascular mortality).

7. What will be the clinical implication in the future of using this tool? How the authors envisage to develop and use such tool? Are they planning to perform a prospective interventional study with which aims?

Reviewer #2: It is a research paper that has been hard work with a large number of patients, but it requires extensive revision in order for the general reader to understand.

I have several questions.

I think you defined DWcp as usual meaning, but what is the ‘prediction of DWcp’ using machine learning?

You said OHcp is the gap between DWcp and the body weight (pre-dialysis?), but what does OHcp mean by machine learning?

You said that you derived a more clinically accurate DW using machine learning, but where is the data?

What new data is included in Group Two?

DWgap Group 2 has mixed definitions. Be sure to indicate.

More detail explanation is needed for the general readers to understand machine learning.

What does the number next to diabetes or HTN on the figure 2? Please indicate the statistical values.

I do not understand what correlation figure 2 represents. The graph showing the correlation seems to be missing.

For the title ‘The factors influencing GapDW’, these factors do not affect GapDW, but are values that depend on the GapDW group.

Maybe larger gap group tend to be in high overhydration status, so that they have large ECW and lower concentrations of materials.

Reviewer #3: Kim et al. have gathered an impressive dataset: BIS-measurements from as many as 1672 hemodialysis patients, undergoing hemodialysis at a single center. The authors stated they would “predict the clinical dry weight, by using machine learning for volume status derived from BIS and clinical information”.

Major comments:

1) I do not understand the definition of the “DW gap”. The authors state that this is “the absolute value of the body weight minus DWBIS from DWCP in units of kg”, but I find this definition unclear. For example, if an 80 kg person has a DWBIS of 77 kg and a DWCP of 78 kg. I am assuming the gap is 1 kg. What happens if an 80 kg person had a DWBIS of 78 kg and a DWCP of 77 – is the gap still 1 kg?

2) The authors state that “in general, dry weight (DW) is defined as the lowest weight a patient on chronic hemodialysis can tolerate [4].” (1) The reference is a 1980 publication by Henderson and clinically outdated. A more appropriate definition is given by Agarwal: (2) (also not the newest paper available, but much more modern than Henderon’s).

3) The authors state that “In this study, the concept of clinically proper DW (DWCP) was used to distinguish it from DWBIS. DWCP was defined as the post-dialysis weight in which the patient had a clinically stable water state (no hypotension during dialysis or edema after dialysis).” Sounds very good, but what was done to ensure this condition in 1672 hemodialysis patients?

4) The authors state: “Additionally, by correcting these influential factors, we attempted to correct the difference between the DWBIS and the clinically appropriately adjusted DW to ultimately predict the correct DW.” A confusing sentence, but even I take it seriously, may I ask how this was done in a retrospective study?

5) The authors state: “A total of 1,672 patients were included in the study.” […] “The study was conducted […] between January 2010 and September 2015.” […] “There are some limitations to this study. First, the DW measurements using the BIS device were performed by a single person [36]”

The reference (REF 36) does not fit here. Can the authors please explain, using a patient flow chart, how exactly that one person was able to perform BIS in all 1,672 patients? Did one person measure 1-2 consecutive patients per day, thus between 300 and 600 a year? How did the authors deal with the effect of time? Did the dry weight protocol change from beginning to end of the retrospective study period?

6) The authors state that “in patients with a GapDW value of more than 2 kg, the proportion of women was high, the age was high, and there were many patients with diabetes (Table 1).” Related to my comment (1), I do not understand how the dry weight gap was defined. Was the clinical dry weight “off target” (too high, if BIS was considered the gold standard) in older women with diabetes?

7) The authors state that “There were 672 (40%) men.” This percentage is very low for a usual hemodialysis center, as the majority of dialysis patients worldwide are men, not women. Can the authors please report the country statistics for Korea, and how their center fits in within their country’s data?

8) The authors state: “It is obvious from this result that, when data with large GapDW are included and as GapDW increases, it is increasingly difficult to predict the target property (OHCP) as a BIS measurement property.” I have no clue what this sentence means.

9) The authors state: “Box plots were used to reveal the distributions of the BIS-based data and the clinical data according to the gap differences based on GapDW. In the box plots, the mean value of ECW showed a tendency to decrease as GapDW increased. Conversely, the overall distribution of hemoglobin, serum total protein, serum albumin, serum creatinine, phosphorus, and serum potassium had lower values as GapDW increased. The ECW value reflects overhydration. As GapDW increases, ECW tends to increase.”

Can the authors please clarify what they mean? In my understanding, the two sentences that I underlined (“In the box plots, the mean value of ECW showed a tendency to decrease as GapDW increased.” versus “As GapDW increases, ECW tends to increase.”) say exactly the opposite of one another.

Overall judgement:

Many of my points raised here above are so major that they unfortunately jeopardize the entire study itself. However, I see an even more fundamental problem with this analysis: If I understand correctly, the authors are using BIS dry weight data to predict clinical dry weight, and the machine learning algorithm is a fancy way of trying to relate one thing (BIS technology) with another (clinical dry weight assessment and management). In my opinion, this undertaking is unfortunately useless, clinically. Many studies from the early years of the BCM have shown that dialysis patients are “not on target”, meaning the clinical dry weight differs from the BIS-derived dry weight. The fact that it is difficult to perform clinical dry weight management is the very reason that BIS is informative, on top of clinical judgement, in the first place. A machine learning algorithm should not be used to identify factors that predict the clinical dry weight, deviating from the BIS-derived dry weight. Instead, the authors can take the classical approach: establish a hypothesis regarding the factors that they feel are important (malnutrition, BMI, sex/gender, malnutrition, inflammation, frailty) and check whether these factors differ between patients who are “on” or “off” the BIS-derived dry weight target. Unfortunately, this analysis will not be new. But in my opinion, the novel machine learning algorithm presented here causes more confusion than it helps the clinician.

Minor:

10) The authors state: “ECW, ICW, and TBW were calculated using a fluid model [23].” The BCM result is actually obtained that way, but this sentence sounds as if the authors had done the calculation by themselves.

References

1. Henderson LW: Symptomatic hypotension during hemodialysis. Kidney Int 1980;17:571-576

2. Agarwal R, Weir MR: Dry-weight: a concept revisited in an effort to avoid medication-directed approaches for blood pressure control in hemodialysis patients. Clin J Am Soc Nephrol 2010;5:1255-1260

6. PLOS authors have the option to publish the peer review history of their article (what does this mean?). If published, this will include your full peer review and any attached files.

Reviewer #1: No

Reviewer #2: No

Reviewer #3: No

---

## [Author Response · Author response to Decision Letter 0]

30 Mar 2021

PLOS ONE

Journal Requirements:

ANS) The manuscripts have been modified according to PLOS one style.

2. To comply with PLOS ONE submission guidelines, in your Methods section, please provide additional information regarding the methodological and statistical analyses performed for the computational tests. For more information on PLOS ONE's expectations for statistical reporting, please see https://journals.plos.org/plosone/s/submission-guidelines.#loc-statistical-reporting.

ANS) We analyzed typical baseline statistics, mean and standard deviation as a characteristics of the data. It is general and clear in statistical analyses. Anyhow, we added more words for more detail in method section, especially, in its subsection, ‘Measurements of clinical parameters’.

ANS) As you indicated, we contain following contents in cover letter. 

Patient’s data contain potentially identifiable or sensitive patient information. The IRB has maintained ethical and legal restrictions on patient data acess. If you want to access individual patient data, you need to get approval from Chungnam National University Hospital IRB (255 Munwharo, Junggu, Daejeon, South Korea, 35015, +82422806781).

ANS) As your recommendation, we corrected that

5.Thank you for stating the following in the Acknowledgments Section of your manuscript:

"This research was also supported by National Institute for Mathematical Sciences (NIMS) grant funded by the Korea government, 2020 (No. NIMS-2020B900000). This research was supported by Basic Science Research Program through the National Research Foundation of Korea (NRF) funded by the Ministry of Education (NRF-2017R1D1AB03035061)."

We note that you have provided c that is not currently declared in your Funding Statement. However, funding information should not appear in the Acknowledgments section or other areas of your manuscript. We will only publish funding information present in the Funding Statement section of the online submission form.

ANS) As your recommendation, we deleted the funding information in Acknowledgments Section. We added the our amended statements within your cover letter. 

Reviewers' comments:

5. Review Comments to the Author

Reviewer #1: Choi and coworkers are submitting a novel approach for adjusting dry weight in hemodialysis patients using machine learning support. For this purpose, they developed a retrospective study using a data set of 1672 hemodialysis (HD) patients in which they compared dry weight defined clinically (clinical proper) to dry weight predicted or estimated from bioimpedance spectroscopy device (BIS). They estimated the gap between these two methods and they clustered such difference in three categories (<1; 1-2;>2 liters). With machine learning support, they identified and weighted factors (clinical data, lab test, body composition from BIS) being associated with each of these groups. They concluded that machine learning support, may help to improve dry weight clinical estimate using BIS and additional factors (clinical data, lab values, body composition) particularly in the lowest fluid overload patient categories. In addition, malnutrition related factors may explain most of gap discrepancy in dry weight estimate.

This novel approach relying on machine learning and artificial intelligence is interesting, and seems appealing as support tool for guiding physicians in managing more precisely dialysis patients. This approach may be used as an example of featuring future of medicine.

Now the study raises several concerns that should be addressed:

1. Use of such machine learning and intelligence artificial tool to identify factors that may influence bad estimate of dry weight such malnutrition is not sufficient to validate this approach.

ANS) Traditional statistical analysis gives many important analytical results. Boxplot, which is one of Exploratory Data Analytics (EDA) is good analytics tool to get interquartile behavior of given data and a pattern in different groups. In this study, we mainly used box plot to identify important factors. However, because of linear property of statistical analysis, it is hard to get hidden nonlinear relations. For the reason, we applied a machine learning approach to find hidden features with its ‘feature importance’ functionality, and then provided its results. Combining statistical analysis and machine learning could get more information and more chance find new factors, especially, important factor for target property, for example, DWCP or OHCP in this study. Because of a little ambiguity, which is the review’s comment, we remedy the statement in the discussion section, as follows. 

 “ In conclusion, machine learning is useful to understand the feature importance of groups according to differences in GapDW. Machine-learning predictive models can be used to establish a clinical diagnosis support system for OHCP.”

 � “ In conclusion, we found the feature importance of groups according to differences in GapDW. Machine-learning predictive models can be helpful to establish OHCP.”

2. It is not clear from reading, how the authors defined the ideal dry weight of patients? Was it the fluid overload estimated from the BIS device that was chosen as target to set suitable or ideal dry weight or the clinical judgement or a combination? That should be defined clearly since it is confusing along the manuscript.

ANS) In this study, the ideal dry weight (clinically proper dry weight (DWCP) in this study) was determined by applying the concepts of several studies that define the existing dry weight [references 1,2,3]. The lowest post-dialysis body weight was set during or after dialysis with no signs or symptoms of overhydration or dehdyration. The ideal dry weight was set by combining the evaluation of the clinical situation while referring to the dry weight predicted through the BIS device. For example, in the case of overhydration, the judgment included peripheral edema, generalized edema, or chest discomfort, or when pleural effusion or pulmonary edema was not observed in chest PA. Hypotension during or after dialysis, musle spasm during dialysis, dizziness It was evaluated based on the presence or absence of triggers. The following content has been added to manuscipt.

Reference 

1) Henderson LW. Symptomatic Hypotension During Hemodialysis. Kidney international. 1980.

2) Jaeger JQ, Mehta RL. Assessment of Dry Weight in Hemodialysis: An Overview. J Am Soc Nephrol. 1999;10(2):392-403.

3) Charra B, Laurent G, Chazot C, Calemard E, Terrat J-C, Vanel T, et al. Clinical Assessment of Dry Weight. Nephrology Dialysis Transplantation. 1996;11(supp2):16-19.

3. If the ideal or suitable dry weight was established on the clinical and/or biologic assessment, therefore the authors should define which criteria and threshold values were used to define the three categories of gap discrepancies.

ANS) There is neither recommended values nor RCT for range of clinical stable dry weight in hemodialysis patients. And, there is no RCT on how much volume change is allowed in hemodialysis patients from an appropriate dry body weight, but when the weight gain between dialysis session is greater than 4-4.5% of the body weight, the risk of ventricular hypertrophy, cardiovascular death, cerebral side effects, all causes mortality is higher (reference 1). For this reason, the U.S. Kidney Foundation generally requires that the weight gain between dialysis session be kept within 2 kg (reference 2). Based on these details, a dry weight error of 2 kg or more was determined as a value with a high possibility of causing side effects due to clinically obvious excessive volume change in hemodialysis patients.

For dry weight errors within 1kg, the BIS measurement error range of volume status is suggested as -1.1~1.1L (referece 3). Based on this information, we set the group of an error of less than 1 kg of dry weight as control. 

1) Bossola M, Pepe G, Vulpio CJJoRN. The frustrating attempt to limit the interdialytic weight gain in patients on chronic hemodialysis: New insights into an old problem. 2018;28(5):293-301.

2) Daugirdas JT, Blake PG, Ing TS. Handbook of dialysis: Lippincott Williams & Wilkins; 2012.

3) Passauer J, Petrov H, Schleser A, Leicht J, Pucalka KJNDT. Evaluation of clinical dry weight assessment in haemodialysis patients using bioimpedance spectroscopy: a cross-sectional study. 2010;25(2):545-551.

4. If the target dry weight was established from BIS measurement then it is easier to understand but still clinical criteria used to define fluid status needs to be listed with their threshold values.

ANS) The criteria for establishing clinical dry weight are the same as those described in the answer to question 2. In order to know the critical value of clinical dry weight, it is necessary to measure the weight which the patient's clinical factors change during dialysis are unstable. However, it is difficult to know the threshold of clinical dry weight. Because of the safety of patients, doctor do not induce the patient to be unstable clinical situation for evaluating range of dry weight. Deliberately lowering or raising dry weight until complications are triggered in order to establish precise criteria for threshold values can be a problem in terms of patient safety. For example, a patient with a weight of 68 kg started dialysis and continue to reduce the weight by 1-2 kg for a dialysis session and find the weight which patient is clinically stable. If the stable dry weight is 64 kg, we do not set the lower dry weight to know the clinical unstable point for patient safety. However, in general, clinical situation remained stable at a range from -0.5kg to 1kg in clinically stable dry weight.

5. From a methodological point of view, it would be also interesting to validate the algorithm that was developed in the assessment of fluid overload within an external cohort of patients.

ANS) This study is a retrospective, a single-center study. It has not been analyzed in an external cohort. We added this point as a limitation of this study. We consider a prospective study including a group of patients from other institutions.

6. Clinical outcomes of these three categories of patients would require further analyses. This is important to value the support of machine learning in reducing intradialytic morbidity (ie, incidence of intradialytic hypotension) and/or improving medium- or long-term morbidity (ie, hospitalization for pulmonary edema) and mortality (all cause or cardiovascular mortality).

ANS) In this study, we focused on prediction of clinical dry weight through machine learning. And the information of the episode of hypotension and mid- to long-term morbidity and mortality during actual dialysis are not collected. Especially we did not permit the information of death and hospitalization of patients from IRB. So, we cannot analysis the intradialytic morbidity and long term mortality. This comment is indicated as a part of the limitation of this research.

7. What will be the clinical implication in the future of using this tool? How the authors envisage to develop and use such tool? Are they planning to perform a prospective interventional study with which aims? 

ANS) Although we did not present it in this study, it is clinically valuable to predict an accurate dry weight because volume status in dialysis patients is already closely related to the patient's prognosis in previous studies.

Several previous studies have addressed limits on dry weight on determining by clinical judgment. In order to compensate for this, there has been an opinion that predicting dry weight using the BIS method. Although BIS based dry weight is useful, but it often incorrectly predicts clinical dry weight. Therefore, it is necessary to find the better prediction tool for clinical stable dry weight. 

In this study, we wanted to know whether a more accurate clinical dry weight could be presented using machine learning based on the various clinical factors and BIS data of the patient. 

Prediction using machine learning will be important because the accuracy increases as the number of data is richer, so it will be important to have a large number of data not only in our institution but also in multi centers. In future prospective studies, the goal is to proceed as a multicenter, large-scale study, and the primary goal is to increase the accuracy of prediction using a large amount of data. In addition, we want to evaluate whether this machine learning based DW prediction improve cardiovascular complications or patient’s survival. 

Reviewer #2: It is a research paper that has been hard work with a large number of patients, but it requires extensive revision in order for the general reader to understand.

I have several questions.

1. I think you defined DWcp as usual meaning, but what is the ‘prediction of DWcp’ using machine learning? 

ANS) The main subject is DWcp, which is easily calculated to OHcp with the body weight. In developing a predictive model in machine learning, it is important to get a better error (small error) rather than accuracy. Sometimes, the good accuracy does not imply an expected good resolution in error. For a good resolution in prediction, it is more efficient to get target factor in small range and decimal point for establishing a predictive model using machine learning. OHcp is the factor in this study. So, we first developed the machine learning predictive model for OHcp rather than DWcp. Hence, we predicted OHcp and then easily calculated DWcp, which is the ‘prediction of DWcp’.

2. You said OHcp is the gap between DWcp and the body weight (pre-dialysis?), but what does OHcp mean by machine learning?

ANS) As we answered in reviewer’s comment 1 just above, we made OHcp the target factor of machine learning rather than DWcp. And then we calculated/recoverd the ‘prediction of DWcp’ with given the body weight and OHcp.

3. You said that you derived a more clinically accurate DW using machine learning, but where is the data?

ANS) Because of the relation among DWcp, OHcp, and the body weight (which is the given input factor), DWcp and OHcp are one-to-one correspondence with the given body weight. Especially, the error of OHcp in prediction is equivalent to that of DWcp in prediction. So, the better (smaller) error of OHcp using machine learning means a more clinically accurate DW. 

4. What new data is included in Group Two?

ANS) Group 2 contains the data of Group 1. So, the new data in Group 2 means 1 kg ≦ GapDW < 2 kg added to group 1. This new data ranges from 1 kg to 2 kg are in Table 1.

5. DWgap Group 2 has mixed definitions. Be sure to indicate.

ANS) we correct the confusing and incorrect expression. Group 2 means <2kg. 

6. More detail explanation is needed for the general readers to understand machine learning.

ANS) As your recommendation, we added the following contents in method section. 

 Since many external sources on machine learning exist, detailed information on the introduction and methodology of machine learning is not included. Briefly, machine learning is a very effective method of learning a given data to predict a specific aim attribute (OHcp in this paper), and using it to predict the corresponding aim attribute when new patients data are inputted. A lot of research is being conducted, and it is widely used in artificial intelligence technology.In this study, we mainly used decision tree-based learning methods (LightGBM, XGBoost, RF) in machine learning predictions. These tree-based methods are more efficient for table type data structure and relatively small data set, especially, RF is better than others for given small data set. Even though deep learning is more famous in machine learning applications, it requires bigger data set for learning model and getting certain prediction accuracy. 

7. What does the number next to diabetes or HTN on the figure 2? Please indicate the statistical values.

ANS) In the box plot, the 1/0 categorical data must be displayed as shown in the figure. Therefore, it is difficult to correct the picture for this part, and instead, the percentage of patients is presented on the graph.

8. I do not understand what correlation figure 2 represents. The graph showing the correlation seems to be missing.

ANS) I removed it because it misrepresented.

9. For the title ‘The factors influencing GapDW’, these factors do not affect GapDW, but are values that depend on the GapDW group. 

ANS) we changed to “ The factors affecting GapDW”

10. Maybe larger gap group tend to be in high overhydration status, so that they have large ECW and lower concentrations of materials. 

Ans) Yes, we agree. Thank you for your comment. 

Reviewer #3: Kim et al. have gathered an impressive dataset: BIS-measurements from as many as 1672 hemodialysis patients, undergoing hemodialysis at a single center. The authors stated they would “predict the clinical dry weight, by using machine learning for volume status derived from BIS and clinical information”.

Major comments:

1) I do not understand the definition of the “DW gap”. The authors state that this is “the absolute value of the body weight minus DWBIS from DWCP in units of kg”, but I find this definition unclear. For example, if an 80 kg person has a DWBIS of 77 kg and a DWCP of 78 kg. I am assuming the gap is 1 kg. What happens if an 80 kg person had a DWBIS of 78 kg and a DWCP of 77 – is the gap still 1 kg?

ANS) As you indicated, the predicted dry weight value by the BIS method may be higher or lower than the clinical dry weight. We did not distinguish theses separately. The focus was mainly on the difference (Gap) between clinical dry weight and BIS based predicted dry weight. We aimed to reduce this gap through machine learning. 

2) The authors state that “in general, dry weight (DW) is defined as the lowest weight a patient on chronic hemodialysis can tolerate [4].” (1) The reference is a 1980 publication by Henderson and clinically outdated. A more appropriate definition is given by Agarwal: (2) (also not the newest paper available, but much more modern than Henderon’s).

ANS) The concept of dry weight proposed by Agarwal is more concrete, but it is difficult to regard it as a completely different concept from the concept proposed by Henderson. The dry weight (DW) defined by Henderson is defined as the lowest weight a patient on chronic hemodialysis can tolerate. The dry weight as defined by Agarwal is the lowest tolerated postdialysis weight achieved via gradual change in postdialysis weight at which there are minimal signs or symptoms of hypovolemia or hypervolemia. As you commented, we modified it in manuscipt. However, since it is almost similar to the concept of dry weight (DW) defined by Henderson, it does not seem necessary to change in the clinically proper dry weight (DWCP) defined in this study. 

3) The authors state that “In this study, the concept of clinically proper DW (DWCP) was used to distinguish it from DWBIS. DWCP was defined as the post-dialysis weight in which the patient had a clinically stable water state (no hypotension during dialysis or edema after dialysis).” Sounds very good, but what was done to ensure this condition in 1672 hemodialysis patients?

ANS) Basically, the hemodialysis medical staffs in this hospital reviews all the symptoms, signs, and events that occurred newly after dialysis immediately before the dialysis day the patient visited. All of the vital sign measured prior to dialysis is checked for stability, and abnormal symptoms or signs are checked through physical examination and history taking such as palpation and auscultation. In addition, the blood pressure during dialysis is basically checked every hour, but when the dry weight is reduced or increased by measuring body composition (for 1-2 weeks), blood pressure is measured at 30-minute intervals during dialysis, and blood pressure fluctuations are large or blood pressure. If there is a deterioration, it is systemized to notify the doctor in charge immediately. In addition, after changing the dry weight, the chest x-ray was checked to check for pulmonary edema or pleural effusion. It was not taken with an exact protocol in relation to the dry weight change date, but it was mainly taken between 1 week and 2 weeks. Depending on the patient, if excessive body water was suspected, continuous chest x-ray was performed every dialysis day.

4) The authors state: “Additionally, by correcting these influential factors, we attempted to correct the difference between the DWBIS and the clinically appropriately adjusted DW to ultimately predict the correct DW.” A confusing sentence, but even I take it seriously, may I ask how this was done in a retrospective study?

ANS) As you indicated, that sentence is misleading. So, we changed the sentence as following, 

Additionally, we evaluated whether a more accurate clinical dry weight could be presented using machine learning based on the various clinical factors and BIS data of the patient.

5) The authors state: “A total of 1,672 patients were included in the study.” […] “The study was conducted […] between January 2010 and September 2015.” […] “There are some limitations to this study. First, the DW measurements using the BIS device were performed by a single person [36]”

◊ The reference (REF 36) does not fit here. Can the authors please explain, using a patient flow chart, how exactly that one person was able to perform BIS in all 1,672 patients? Did one person measure 1-2 consecutive patients per day, thus between 300 and 600 a year? How did the authors deal with the effect of time? Did the dry weight protocol change from beginning to end of the retrospective study period?

ANS) For Reference 36 insertion, it is a duplicate insertion. Thanks for the right point. I deleted it.

In hospitals where the research was conducted, specialized nurses trained to perform outpatient examinations, etc. reside in the internal medicine department. During the study period, one professional nurse performed the same task, and the dedicated nurse was in June 2020. Dedicated nurses worked daily from 9am to 5pm, Monday through Friday. The number of patients examined was 0-7 per day, which varied from day to day. The examination was not performed while the nurse was absent, and the period of absence was very short, so there were no significant restrictions on the performance of the examination.

Outpatient dialysis patients visited the internal medicine outpatient clinic before dialysis on the day of dialysis and were instructed to measure dry weight through the BIS device, and dialysis was performed immediately after the dry weight measurement.

Dry weight measurement using the BIS device used the same protocol.

6) The authors state that “in patients with a GapDW value of more than 2 kg, the proportion of women was high, the age was high, and there were many patients with diabetes (Table 1).” Related to my comment (1), I do not understand how the dry weight gap was defined. Was the clinical dry weight “off target” (too high, if BIS was considered the gold standard) in older women with diabetes?

ANS) BIS measurement method is not Gold Standard. It should be considered that the clinical dry weight is the gold standard. Although the clinical dry weight is gold standard, finding this value actually takes a lot of time and trial and error because it means that the patient's condition is stable while continuing to dialysis. On the other hand, the BIS method is an easy method because you only have to invest about 10 minutes to measure it, and there is a relatively accurate part in evaluating the patient's hydration status.

However, there are cases where the BIS method is not well suited for predicting the actual clinical dry weight, and in this study, it means that the frequency of patients who do not fit well in the case of elderly and female diabetes in this study is high. The fact that the reviewers did not understand the definition of GAPDW may have determined that the BIS method's prediction of water status is inconsistent due to a mixture of overestimating or underestimating water status than the actual clinical status. However, in this study, we did not focus on the over-underestimation of the patient's hydration status, but whether the BIS method measurement can reduce excessive deviation from the actual clinical dry weight value through machine learning. And we focused on the characteristics of people who are excessively deviated. 

This means that elderly and diabetic women often deviate from the actual dry weight value. 

7) The authors state that “There were 672 (40%) men.” This percentage is very low for a usual hemodialysis center, as the majority of dialysis patients worldwide are men, not women. Can the authors please report the country statistics for Korea, and how their center fits in within their country’s data?

ANS) According to a 2018 report from the Insan Memorial Dialysis Registry (ESRD Registry Committee) collected by the Korean Society of Nephrology, the total number of hemodialysis patients was 77,617, of which 59% were male. In this center, there are variations every year, but the proportion of males in outpatient hemodialysis clinics is around 50%. However, if ward dialysis patients account for 35-45% of the total number of outpatients, it is considered to have a similar value to the Korean Society of Nephrology. In addition, the low proportion of men in this study is thought to have been derived because the proportion of men removed from the exclusion criteria was high, and the data were not separately adjusted to the gender ratio. This study is a single-center study, and we believe that this representation problem can be supplemented when multi-center studies are conducted in the future.

8) The authors state: “It is obvious from this result that, when data with large GapDW are included and as GapDW increases, it is increasingly difficult to predict the target property (OHCP) as a BIS measurement property.” I have no clue what this sentence means.

ANS) It can be seen that the probability of prediction for the target attribute OHcp (Table 2) by utilizing machine learning methods is probabilistically, the larger the GapDW is in the group containing the larger value.

I used the expression that DWgap is more difficult because the absolute error is increasing in the group with large DWgap, and the large group is increasing in the group with large DWgap.

9) The authors state: “Box plots were used to reveal the distributions of the BIS-based data and the clinical data according to the gap differences based on GapDW. In the box plots, the mean value of ECW showed a tendency to decrease as GapDW increased. Conversely, the overall distribution of hemoglobin, serum total protein, serum albumin, serum creatinine, phosphorus, and serum potassium had lower values as GapDW increased. The ECW value reflects overhydration. As GapDW increases, ECW tends to increase.”

◊ Can the authors please clarify what they mean? In my understanding, the two sentences that I underlined (“In the box plots, the mean value of ECW showed a tendency to decrease as GapDW increased.” versus “As GapDW increases, ECW tends to increase.”) say exactly the opposite of one another.

ANS)As GapDW increases, ECW tends to increase. Modified in Manuscipt.

Overall judgement:

Many of my points raised here above are so major that they unfortunately jeopardize the entire study itself. However, I see an even more fundamental problem with this analysis: If I understand correctly, the authors are using BIS dry weight data to predict clinical dry weight, and the machine learning algorithm is a fancy way of trying to relate one thing (BIS technology) with another (clinical dry weight assessment and management). In my opinion, this undertaking is unfortunately useless, clinically. Many studies from the early years of the BCM have shown that dialysis patients are “not on target”, meaning the clinical dry weight differs from the BIS-derived dry weight. The fact that it is difficult to perform clinical dry weight management is the very reason that BIS is informative, on top of clinical judgement, in the first place. A machine learning algorithm should not be used to identify factors that predict the clinical dry weight, deviating from the BIS-derived dry weight. Instead, the authors can take the classical approach: establish a hypothesis regarding the factors that they feel are important (malnutrition, BMI, sex/gender, malnutrition, inflammation, frailty) and check whether these factors differ between patients who are “on” or “off” the BIS-derived dry weight target. Unfortunately, this analysis will not be new. But in my opinion, the novel machine learning algorithm presented here causes more confusion than it helps the clinician.

Minor:

10) The authors state: “ECW, ICW, and TBW were calculated using a fluid model [23].” The BCM result is actually obtained that way, but this sentence sounds as if the authors had done the calculation by themselves.

ANS) The following sentences have been deleted to eliminate misunderstandings.

References

1. Henderson LW: Symptomatic hypotension during hemodialysis. Kidney Int 1980;17:571-576

2. Agarwal R, Weir MR: Dry-weight: a concept revisited in an effort to avoid medication-directed approaches for blood pressure control in hemodialysis patients. Clin J Am Soc Nephrol 2010;5:1255-1260

---

## [Editor Report · Decision Letter 1]

7 Apr 2021

A novel approach to dry weight adjustments for dialysis patients using machine learning

PONE-D-20-40081R1

Dear Dr. Choi,

We’re pleased to inform you that your manuscript has been judged scientifically suitable for publication and will be formally accepted for publication once it meets all outstanding technical requirements.

Kind regards,

Bhagwan Dass, MD

Academic Editor

PLOS ONE
---

## [Editor Report · Acceptance letter]

12 Apr 2021

PONE-D-20-40081R1 

A novel approach to dry weight adjustments for dialysis patients using machine learning 

Dear Dr. Choi:

I'm pleased to inform you that your manuscript has been deemed suitable for publication in PLOS ONE. Congratulations! Your manuscript is now with our production department. 

Kind regards, 

on behalf of

Dr. Bhagwan Dass 

Academic Editor

PLOS ONE